# Lonicera Caerulea Juice Alleviates Alcoholic Liver Disease by Regulating Intestinal Flora and the FXR-FGF15 Signaling Pathway

**DOI:** 10.3390/nu15184025

**Published:** 2023-09-17

**Authors:** Baixi Zhang, Lijuan Niu, Xinwen Huang

**Affiliations:** National Engineering Research Center for Functional Food, School of Food Science and Technology, Jiangnan University, Wuxi 214122, China; 6200113070@stu.jiangnan.edu.cn (L.N.); 6200112030@stu.jiangnan.edu.cn (X.H.)

**Keywords:** *Lonicera caerulea*, alcoholic liver damage, gut microbiota, FXR–FGF15 signaling pathways, short-chain fatty acid

## Abstract

Alcoholic liver disease (ALD) is a growing public health issue with high financial, social, and medical costs. *Lonicera caerulea*, which is rich in polyphenolic compounds, has been shown to exert anti-oxidative and anti–inflammatory effects. This study aimed to explore the effects and mechanisms of concentrated *Lonicera caerulea* juice (LCJ) on ALD in mice. ALD was established in mice via gradient alcohol feeding for 30 days. The mice in the experimental group were given LCJ by gavage. The reduction of aspartate transaminase (AST) and alanine transaminase (ALT) in the serum of mice indicated that LCJ has a liver-protective effect. LCJ improved the expression of AMPK, PPARα, and CPT1b in ALD mice to reduce the liver lipid content. Additionally, LCJ increased the expression of farnesoid X receptor (FXR), fibroblast growth factor 15 (FGF15), and fibroblast growth factor receptor 4 (FGFR4), which lowers the expression of cytochrome P450 7A1 (CYP7A1) and lessens bile acid deposition in the liver. In mice, LCJ improved the intestinal barrier by upregulating the expression of mucins and tight junction proteins in the small intestine. Moreover, it accelerated the restoration of microbial homeostasis in both the large and small intestines and increased short–chain fatty acids in the cecum. In conclusion, LCJ alleviates ALD by reducing liver and serum lipid accumulation and modulating the FXR–FGF15 signaling pathway mediated by gut microbes.

## 1. Introduction

The chronic daily intake of a specified quantity of alcohol can lead to the development of alcoholic liver disease (ALD), which comprises a spectrum of ailments ranging from reversible hepatic steatosis to acute alcoholic hepatitis, chronic fibrosis, cirrhosis, and hepatocellular carcinoma [1]. ALD is a leading cause of morbidity and mortality worldwide [2,3]. Upon entering the body, the majority of alcohol is promptly transported to the intestines, where over 90% of it is absorbed into the bloodstream [4]. The first-pass metabolism of alcohol occurs in the stomach; however, compared with the liver, the stomach has a lower amount of alcohol-metabolizing enzymes. Thus, the liver serves as the primary organ responsible for alcohol metabolism and breakdown within the body [5]. The intestines and liver are, thus, the main organs that are damaged by alcohol. The pathogenesis of ALD is complex, and the existing treatment methods of abstinence, diet, and drug therapy for ALD have some drawbacks [6,7,8]. Therefore, exploring the pathogenesis of ALD to identify a suitable treatment for the disease is essential.

Prior research has demonstrated that edible plants and their bioactive compounds, such as polyphenols, can safeguard against ALD by modulating the gut microbiota [9,10]. By inhibiting the expression of genes associated with lipid production and encouraging the expression of genes linked to lipid breakdown in ALD mice, 200 mg/kg of blueberry polyphenol extract can help to alleviate liver damage [11]. Grape–leaf extract—which is rich in phenolic compounds (250–500 mg/kg)—was orally given to ethanol-induced rats for 12 days, which reduced liver injury by improving antioxidant activity and inhibiting nuclear factor kappa B p65 and pro–inflammatory cytokines (tumor necrosis factor-alpha) [12]. The honeysuckle plant *Lonicera caerulea* (LC) is classified as a “homology of medicine and food” fruit due to its high concentration of bioactive compounds [13]. High levels of phenolic chemicals were found in LC, including catechin, procyanidin, chlorogenic acid, anthocyanin 3–glucoside (C3G), etc. [14]. The total phenolic compound content of LC reached 1140.06 mg/100 g FW, which is much higher than that of blueberries (94.60–137.74 mg/100 g FW) [15,16]. In addition, LC polyphenol extract alleviates non-alcoholic fatty liver by inhibiting proinflammatory cytokine production and lipid peroxidation. [17]. LC polyphenols can regulate the intestinal epithelial barrier function and microbiome of Sprague Dawley rats on a high-fat diet to inhibit fat absorption [18].

Herein, we hypothesized that LC—which is high in polyphenols—can alleviate ALD. We developed a mouse model of ALD to determine if *Lonicera caerulea* juice (LCJ) can alleviate the disease and examine potential mechanisms. In addition to promoting the thorough development and utilization of LC resources, our results may also provide a new reference for the treatment of ALD.

## 2. Materials and Methods

### 2.1. Preparation of LCJ

Pure water and *Lonicera caerulea* were combined in a colloid mill at a ratio of 1:1 (*w*/*v*), and the pulp was processed repeatedly. The juice was collected after the pulp had been centrifuged at 3000 rpm for 30 min. Finally, the juice was evaporated by rotation into two types of LCJ, each with a total phenol content of 9 mg/mL and 18 mg/mL. The *Lonicera caerulea* was provided by the Heilongjiang Fengran Agricultural Group Company. The company certified that the *Lonicera caerulea* L. was the “Beilei” variety, belonging to the *Lonicerae* genus of the *Caprifoliaceae* family. 

### 2.2. Determination of Phenolic Components in LCJ by Liquid Chromatography–Mass Spectrometry

LCJ with a total phenolic content of 18 mg/mL was diluted by a factor of four, and then about 100 μL of the sample was transferred to an Eppendorf tube. After the addition of 300 μL of extract solution (methanol), samples were vortexed for 30 s, sonicated for 10 min in an ice–water bath, and incubated for 2 h at −20 °C to precipitate the proteins. Then, the sample was centrifuged at 12,000 rpm for 15 min at 4 °C. The resulting supernatant was transferred to a fresh glass vial for analysis. A triple quadrupole compound linear ion trap liquid-mass coupling instrument (Applied Biosystems, Inc, CA, USA) was used to detect phenols in LCJ [19]. Chromatographic separations were performed in a BEH C18 column (2.1 × 100 mm i.d., 1.7 µm particle diameter). Solvent A was water (0.1% formic acid aqueous solution), and solvent B was acetonitrile. The column temperature was kept at 40 °C. The program was performed as follows: 0–10 min, 98% A and 2% B; 10–12 min, 2% A and 98% B; and 12–15 min, 98% A and 2% B at a flow rate of 0.30 mL/min. And the injection volume was 1 µL.

### 2.3. Animal Experiments

It was reported that the chronic plus––binge ethanol challenge induced a greater degree of adipose tissue inflammation and liver injury in female mice than in male mice [20,21]. Therefore, the choice of female mice as the research subjects in our experiment was advantageous for successful modeling of alcoholic liver disease in mice as well as for future population studies on liver damage in the female alcoholic population. Thirty–six female mice of the C57BL/6N background (10 weeks old, weighing 20 ± 2 g) were purchased from Beijing Vital River Laboratory Animal Technology (Beijing, China) and housed at a temperature of 22 ± 2 °C and a humidity of 50 ± 10% under a 12 h light/dark cycle in a specific pathogen-free animal facility. All animal experiments were examined and authorized by the Laboratory Animal Ethics Committee of Jiangnan University (Ethical approval number: JN. No20220615c0450803223). Mice had access to water and a normal diet for five days of the adaptation period.

It has been proven that 200 mg/kg of blueberry polyphenol and grape leaf extract containing 250 mg/kg of polyphenol can effectively relieve ALD [11,12]. Since phenolic compounds are the main active substances in LC, the test groups were divided into two groups according to their total phenol intake. After the adaptation period, mice were randomly divided into 4 groups with 9 mice in each group: the control group (CG), the alcohol group (AG), the low-dose LCJ group (LG) (total phenol intake: 150 mg/kg), and the high-dose LCJ group (HG) (total phenol intake: 300 mg/kg). The dosage of concentrated fruit juice given to mice was 300 mg/kg and 150 mg/kg. This dose was realistic, as it is equivalent to 100 mL of LCJ (the total phenol concentration is 9 mg/mL or 18 mg/mL) per day for humans.

To establish a model of chronic intake and single binge drinking, all mice were given the Lieber-DeCarli control liquid diet for the first 5 days. The model group and the test group were fed with gradient alcohol adaptation mode for 9 days, which was 1.7% (*w*/*v*) alcohol for 3 days, 2.5% (*w*/*v*) alcohol for 3 days, and 3.3% (*w*/*v*) alcohol for 3 days. Then, they were given a Lieber-DeCarli diet containing 5% (*w*/*v*) alcohol for 16 days. On the last day, the experimental group was given a concentration of 31.5% (vol/vol) ethanol solution by gavage, and the volume of the gavage (μL) = body weight (g) × 20. The control group was administered with an isocaloric maltodextrin solution. Mice were sacrificed nine hours after intragastric administration. 

### 2.4. Histopathological Analysis

Specimens of the live large intestine and small intestine were collected for histological analysis. They were fixed in 4% paraformaldehyde/phosphate-buffered saline (PBS) (*v*/*v*) and embedded in paraffin. Samples were cut into small sections for staining with hematoxylin and eosin (HE) and oil red O; sections were then examined using a microscope. According to Osho et al. [22], new large intestine and small intestine tissues were stained with Alcian blue and periodic acid–Schiff reagent for 30 min for goblet cells. Afterward, samples were rinsed in tap water for 10 min, exposed to periodic acid (5 g/L) for 5 min, rinsed in lukewarm water for 10 min, and then counter–stained with Coleman’s Schiff reagent.

### 2.5. Biochemical Assays of Serum and Tissue

Serum was extracted from blood samples by centrifugation at 4000 g for 15 min at 4 °C after being maintained at 4 °C for 2 h. Subsequently, serum levels of cholesterol (TC), alanine transaminase (ALT), aspartate transaminase (AST), triglycerides (TG), and total bile acid (TBA) were evaluated using a BK–400 automatic biochemical analyzer. In addition, the kits of serum ALT, AST, TG, TC, and TBA were purchased from Nanjing Jiancheng Biological Engineering Research Institute (Nanjing, China). Lipopolysaccharides (LPS) ELISA kits were purchased from SenBeiJia Biological Technology Co., Ltd. (Nanjing, China).

A liver sample weighing 0.1 g was homogenized using a high-speed homogenizer in 0.9 mL of physiological saline to produce a liver homogenate. The supernatants were collected by centrifugation at 3000 g for 15 min at 4 °C. The levels of TC, TG, and TBA in the liver were determined using commercial kits obtained from Nanjing Jiancheng Biological Engineering Research Institute (Nanjing, China). Fecal bile acid levels were detected according to a method previously described by Guo et al. [23].

### 2.6. Gut Microbiota Analysis 

All the contents of the large and small intestines were scraped and stirred well. Samples were then divided into 6 groups: SCG, small intestine in the control group; SAG, small intestine in the alcohol group; SHG, small intestine in the high–dose LCJ group; LCG, large intestine in the control group; LAG, large intestine in the alcohol group; LHG, large intestine in the high–dose LCJ group. The total DNA of microorganisms in intestine contents was extracted using a fecal DNA extraction kit (MP Biomedicals, Santa Ana, CA, USA). DNA was successfully separated by agarose gel electrophoresis.

The highly variable region (V3–V4) of bacterial 16S ribosomal DNA was used for polymerase chain reaction (PCR) amplification. The primers were 341F (5′-ACTCCTACGGGAGGCAGCAG-3′) and 806R (5′-GGACTACHVGGGTWTCTAAT-3′). According to the instructions of Nanjing Vazyme Biotech Co., Ltd. (Nanjing, China) reagent kit, a PCR reaction system was prepared by taking a qualified genomic DNA sample of 30 ng and corresponding fusion primers. The PCR products were purified according to the instructions of the AxyPrep DNA Gel Extraction Kit (Axygen company, CA, USA) reagent kit. The Agencourt AMPure XP (Beckman CoulterTM, Brea, CA, USA) magnetic beads were used to dissolve and purify the PCR amplification products before labeling them to finish the library construction. An Agilent 2100 bioanalyzer (Agilent, Santa Clara, CA, USA) was used to analyze libraries for fragment range and concentration. Libraries that passed the test were selected for sequencing on a HiSeq platform (BGI Genomics Co., Ltd., Shenzhen, China) according to insert sizes. The data were filtered, and the remaining high-quality, clean data were used in post-analyses. Sequence splicing was performed using FLASH software (Fast Length Adjustment of Short reads, v1.2.11), which assembles pairs of reads from double-end sequences into a single sequence using overlap relationships to obtain tags for high–variation regions. Amplicon sequence variants (ASVs) were obtained, compared with the database, and annotated with the species using the DADA2 (divisive amplicon denoising algorithm) method in QIIME2. Sample species complexity analysis, intergroup species variation analysis, association analysis, and functional prediction were carried out based on the results of OTU and annotation. The BGI platform made the bioinformatics analytic techniques available for free online.

### 2.7. Determination of Short Chain Fatty Acids (SCFA) in Cecal Contents

About 0.09 g of fecal content homogenate was added to 0.5 mL of saturated sodium chloride solution precooled to 4 °C. Then, 10 µL of a 10% (*v*/*v*) sulfuric acid solution precooled at 4 °C was added to extract the fatty acids, followed by the addition of 1 mL of anhydrous ether (containing 1 mmol/L of internal standard 2–ethylbutyric acid) precooled at 4 °C. The whirlpool was then shaken for 30 s. Samples were centrifuged at 3000 rpm for 15 min at 4 °C. Finally, SCFAs in the supernatant were analyzed using gas chromatography–mass spectrometry. [24].

### 2.8. Real-Time Quantitative PCR

From the liver, total RNA was extracted, and 1 μg of this RNA underwent reverse transcription to produce cDNA. Using SYBR Green qPCR Master Mix, quantitative RT-PCR was carried out on a CFX96 Touch real–time PCR machine. These analyses were carried out with the corresponding commercial kits from Nanjing Vazyme Biotech Co., Ltd. (Nanjing, China). The expression level of the liver gene was normalized to the β-actin gene, while the expression level of the small intestine gene was normalized to the GAPDH gene and calculated utilizing the 2^−∆∆Ct^ method. Primer sequences are presented in Appendix A.

### 2.9. Western Blotting

Three mice in each group were selected for the Western blot assay. The tissues were lysed with RIPA buffer (1 mL) containing proteinase inhibitors (Beyotime, Shanghai, China). The homogenate was centrifuged at 3000 rpm for 5 min at 4 °C. The protein content was determined by the BCA protein concentration assay kit (Beyotime, Shanghai, China) and diluted to a suitable concentration for the experiment.

The protein sample from each mouse was separated by electrophoresis in SDS-PAGE (Amresco, Solon, OH, USA) and transferred onto a polyvinylidene difluoride (PVDF) membrane (Millipore, Schwalbach, Germany). The membrane was incubated with only one primary antibody every time, then incubated with the secondary antibody and detected. Then, all the detection reagents from the blot on the same membrane are removed and reprobed with the other primary antibody for the detection of the second target protein. Specific operations are as follows: Equal quantities of protein were separated by electrophoresis in SDS–PAGE and then deposited onto PVDF membranes. After the membranes had been blocked with 5% nonfat dry milk, the corresponding primary antibodies were incubated on the membranes overnight at 4 °C. After being washed with Tris–buffered saline with Tween, followed by incubation with secondary antibodies for 1 h at room temperature. Both a luminescence imaging workstation (Bio–Rad Laboratories, Inc., CA, USA) and an ECL chemiluminescence kit (Beyotime, Beijing, China) were utilized to create the target protein. When multiple exposures of the same PVDF membrane were required, Western blot stripping buffer was shaken at room temperature for 45 min and washed with TBST 3 times. The subsequent steps were the same as before. Image Pro Plus 6.0 software was used to analyze the optical density value (Tanon Science & Technology Co., Ltd., Shanghai, China). The following antibodies were used: MUC4 (1:100, Santa Cruz), CYP7A1 (1:5000, Santa Cruz), FXR (1:1000, Cell Signaling Technology), MUC2 (1:1000, Abcam), FGFR4 (1:1000, Abcam), ZO-1 (1:1000, Abcam), Claudin-1 (1:1000, Abcam), FGF15 (1:1000, Abcam), GAPDH (1:1000, Abcam), β-actin (1:1000, Abcam), AMPK (1:5000, Abcam), CPT1b (1:1000, Abcam), PPARα (1:1000, Abcam).

### 2.10. Statistical Analysis

Statistical analysis was performed using GraphPad Prism 8.0 software and SPSS 25.0 software. All the data in this study were expressed as the mean ± standard deviation. The statistical significance of the results was analyzed by one-way analysis of variance (ANOVA) based on Duncan’s multiple range test using SPSS. The significance level in the analyses was considered *p* < 0.05.

## 3. Results

### 3.1. The Content of the Phenolic Components in LCJ

The content of 16 phenolic substances in LCJ was determined (Appendix A and Appendix A), which was consistent with the literature reports on the identification of LC components [14]. The highest content in LCJ was cyanidin-3-O-glucoside (C3G), whose content was 394.57 ± 0.06 mg/100 mL. Among the detected chemicals, many have not been quantitatively reported in LC, such as cyanidin–3–galactoside, phlorizin, neodiosmin, naringenin, astragalin, and 7-hydroxycoumarin. Since cyanidin–3–galactoside is an isomer of C3G, its quantitative analysis may have been neglected in previous studies. The content of cyanidin–3–galactoside in LCJ was 275.48 ± 0.22 mg/100 mL, which is the same order of magnitude as that of C3G. In addition, the contents of chlorogenic acid, quercetin, and isochlorogenic acid C were relatively high: 96.44 ± 0.02 mg/100 mL and 16.77 ± 0.31 mg/100 mL, respectively.

### 3.2. Effects of LCJ Intervention on Body Growth Performance and Hepatocyte Damage in ALD Mice

Figure 1a shows the changes in the BW. The BW of the AG group was lower than that of the CG group from the beginning, suggesting that prolonged alcohol intake can inhibit normal growth in mice. The BW of mice in the LG and HG groups was positively correlated with the dose of LCJ intragastric administration, which was higher than that in the AG group, but there was no significant correlation (*p* > 0.05). These results indicated that LCJ could increase the BW of ALD mice. Furthermore, alcohol consumption did not induce any notable alteration in liver weight, and there was no significant difference in liver weight observed among any of the groups (Figure 1b). The liver index of mice treated with ethanol was much higher than that of control mice, and a low dose of LCJ did not significantly lower this proportion, which was significantly lowered by a high dose of LCJ (Figure 1c). These findings support the notion that drinking in this manner may negatively impact the BW and liver index, but high doses of LCJ can alleviate this impact. In addition, alcohol increased serum AST, ALT, and AST/ALT ratios (Figure 1d–f), which is consistent with the results of Wu et al. [25], suggesting that this type of alcohol intake can lead to liver cell damage in ALD mice. Nevertheless, the administration of high–dose LCJ orally resulted in a substantial reduction in the atypical increases of ALT and AST serum activity as well as the AST/ALT ratio induced by alcohol, indicating the hepatoprotective potential of LCJ in ALD mice.

### 3.3. LCJ Improved Liver Fat Accumulation in ALD Mice

The HE staining study revealed that chronic alcohol use resulted in hepatocyte destruction and liver cell steatosis in AG groups (Figure 1g). However, both the LG and HG groups had decreased hepatic steatosis and hepatocyte injury when compared to the model group, and the impact was favorably correlated with dose. LCJ can lower hepatic lipid buildup in ALD mice, according to oil–red O staining (Figure 1h). The increase of serum TC and TG as well as liver TG and TC in AG group mice was brought on by continuous alcohol feeding, which was consistent with the results obtained by Zhou et al. in the alcoholic fatty liver model group [26]. High–dose LCJ could reduce the accumulation of blood lipids and liver lipids in ALD mice (Figure 1i–l). According to the biochemical indicators of lipids in the serum and liver, only high–dose LCJ significantly alleviates ALD in mice. As a result, the subsequent investigation of the effects of LCJ on the expression of substances involved in lipid synthesis and metabolism pathways using qPCR and Western blot methods was limited to the CG group, AG group, and HG group. We found that high– dose LCJ can promote the β oxidation of fatty acids by increasing the protein expression of AMPK, PPARα, and CPT1b in the liver of ALD mice (Figure 2a–c), thereby reducing the accumulation of TG and TC in the liver of ALD mice.

### 3.4. LCJ Alleviates Hepatic BA Deposition in ALD Mice by Regulating the FXR-FGF15 Axis

Compared with the CG group, TBA levels in the liver and serum of the AG group were significantly higher, whereas the TBA levels in the feces were significantly lower. However, high–dose LCJ reduced TBA in the liver and serum and increased TBA in the feces of ALD mice, while low–dose LCJ had no significant effect on TBA in the liver, blood, or feces (Figure 2d–f). Therefore, we determined the content of BA metabolization-related genes in the liver and small intestine of the CG, AG, and HG groups by RT–qPCR and Western blotting. Alcohol reduced the levels of fibroblast growth factor receptor 4 (FGFR4) in the liver and fibroblast growth factor 15 (FGF15) in the small intestine while raising the levels of cytochrome P450 7A1 (CYP7A1) in the liver of the AG group. High doses of LCJ, however, restored these proteins to levels comparable to those in the CG group in ALD mice (Figure 2a and Figure 3c).

### 3.5. LCJ Improves the Intestinal Barrier and Reduces LPS Entry into the Bloodstream in ALD Mice

The results of HE staining and Alcian blue–periodic acid–Schiff (AB-PAS) staining showed that the crypt structure of the small intestinal tissue in the AG group was severely damaged, with a large number of cell deaths and reduced mucin content. Alcohol had little effect on the mucin content and large intestine anatomy. This may be because alcohol is mainly absorbed and metabolized in the small intestine [27]. The effects of alcohol on intestinal mucus and barrier function were consistent with previous reports [28,29]. The structure and mucin content of the small and large intestines in the HG group, however, were restored to levels comparable to those in the CG group following the administration of LCJ (Figure 3a,b). Intestinal permeability can be reflected in serum LPS [27]. Alcohol intake led to elevated blood levels of LPS in the AG group. A high dose of LCJ reduced the blood LPS content of ALD mice (Figure 3c). We believe that low-dose LCJ cannot significantly alleviate intestinal barrier damage in ALD mice, whereas high-dose LCJ can significantly improve intestinal barrier damage in ALD mice based on the results of HE and AB-PAS staining as well as changes in serum LPS. Therefore, we investigated the mechanism by which LCJ alleviates alcohol-induced intestinal barrier damage in mice, focusing on CG, AG, and HG group mice.

The results showed that high-dose LCJ increased the expression of tight junction proteins (ZO–1, claudin–1) and mucins (MUC2 and MUC4) in the small intestinal tract of ALD mice (Figure 3d–f). The results of intestinal pathological section staining supported these findings. These data imply that LCJ supplementation to improve intestinal barrier dysfunction may be an effective treatment for alcoholic liver injury.

### 3.6. LCJ Increase the Level of SCFAs in the Cecum of ALD Mice

Based on the biochemical indicators of the liver and serum of mice, as well as the intestinal barrier, intestinal mucin, and bile acid indicators of secondary metabolites of the microbiota, we believe that only high–dose LCJ can improve ALD in mice. Therefore, our study is limited to the CG, AG, and HG groups regarding the effect of LCJ on the intestinal microbiota and its metabolites, short–chain fatty acids, in ALD mice.

SCFAs are acidic byproducts of the fermentation of difficult-to-digest carbohydrates by bacteria in the gut, which strengthen the intestinal barrier [30]. Ethyl alcohol (EtOH) liquid diet caused a significant drop in the amounts of acetic acid, propionic acid, butyric acid, valeric acid, and total SCAFs in the cecum of the AG group (Figure 4a–g). In addition, alcohol consumption has been shown to reduce SCAFs in the cecum of animals [31,32]. Although not statistically significant, ethanol increased isovaleric acid in the AG group. Acetic acid, propionic acid, butyric acid, valeric acid, and total SCAF concentrations in the cecum of mice were significantly higher after administration of high doses of LCJ, while isovaleric acid contents were significantly lower.

### 3.7. LCJ Mediates Alcohol-Induced Intestinal Flora Disturbance in the Large and Small Intestine

As shown in Figure 5a, the dilution curves of each group tended to be smooth, indicating that there was a sufficient amount of sequencing data. Alcohol consumption had minimal effects on the species richness and homogeneity of the small and large intestines in ALD mice. Supplementation with higher doses of LCJ significantly reversed these effects (Figure 5b).

The Sobs, Chao, and Ace indexes describe community richness. The Shannon and Simpson indices describe community α diversity (Figure 5c–g). Our data showed that alcohol did not change the diversity of coliforms, but it increased their richness. Besides, alcohol reduced the richness and diversity of microorganisms in the small intestine. LCJ decreased the intestinal microbial richness and significantly increased the intestinal microbial richness and diversity of ALD mice. The findings suggested that LCJ exerted distinct regulatory effects on the α diversity of intestinal flora in ALD mice; however, ultimately, it facilitated the restoration of α diversity of intestinal flora in ALD mice, which was consistent with that in the CG group. Previous studies found that alcohol reduced the Shannon index of the cecum [33]. Xu et al. found that alcohol intake increased the α diversity of mouse feces [34]. This may be because the intestine is highly dynamic; small and large intestines are regulated differently under external influences to maintain overall intestinal homeostasis [35].

β diversity analysis was used to show differences in the microbial community composition among different groups of samples (Figure 6a–e). The large and small intestines of normal mice had different flora structures. The intestinal flora structure in the AG group demonstrated significant alterations in both the small and large intestines compared to that of the CG group. These observations align with the findings of Cao et al.’s study [36]; however, the effect of alcohol on the intestinal flora structure was greater in the small intestine. The structure of the intestinal flora in the small intestine in the HG group recovered to resemble that in the CG group following the administration of high-dose LCJ. Although the intestinal flora of the large intestine in the HG group did not recover to resemble that of the CG group, it was also distinct from that of the AG group. 

*Bacteroidetes*, *Verrucomicrobia*, *Actinobacteria*, *Proteobacteria*, and *Firmicutes* were the dominant phyla in the small and large intestinal contents of normal mice (Figure 7a–d). Alcohol increased the relative abundance of Proteobacteria in the small intestine and decreased the relative abundance of *Firmicutes* and *Bacteroidetes* in the small intestine and *Staphylococcus* in the large intestine. LCJ supplementation restored these levels to those in the CG groups (Figure 7e).

The small intestine had 16 key genera, while the large intestine contained 14 key genera (Figure 7f–i). Alcohol intervention considerably increased the relative abundance of *Staphylococcus* and *Proteus* in the small intestine while significantly decreasing the relative abundance of *Allobaculum* and *Akkermansia*. In the large intestine, alcohol intervention increased the relative abundance of *Parabacteroides*, *Oscillospira*, and *Proteus* and decreased the relative abundance of *Akkermansia*. LCJ administration significantly reversed these intestinal microbiota alterations in ALD mice, restoring them to a level similar to that of the CG group (Figure 7j,k).

To explore the effect of LCJ on the microbial structure of the large and small intestines of mice in each group, microbial markers in the large and small intestines of mice in each group were determined using linear discriminant analysis (LDA) effect size (LEfSe) analysis and an LDA analysis score greater than 4. Results are shown in Figure 8a–d. The biomarkers in the small intestine of the AG group included *Enterococcus*, *Prauseria*, and *Proteus*, whereas those in the large intestine were *Proteus*, *Ricerysipelorichi*, *Clostridium*, *Streptococcus*, and *Parabacteroides*. The biomarkers in the small intestine of HG mice included *Allobaculum*, *Akkermansia*, *Bilophila*, and *Sutterella*, whereas those in the large intestine were *Defluviitalea*, *Bacteroides*, *Akkermansia*, *Bilophila*, and *Alistipes*.

## 4. Discussion

ALD is associated with high morbidity and mortality worldwide, which calls for investigations into its pathogenesis to find better treatments. Alcohol consumption can disrupt the composition of the gut microbiota, and resolving the disruption of the gut microbiota caused by alcohol can alleviate ALD. Previous studies have shown that plant natural products and polyphenolic compounds exert beneficial effects on several metabolic disorders associated with ALD via the gut–liver axis [35]. LC is rich in polyphenols such as C3G and can ameliorate non-alcoholic fatty liver disease. We speculate that LC can improve ALD. Meanwhile, the relationship between the modulating effects of LC on the gut microbiota and its protective effects on ALD has not been well studied. Interestingly, LC reduced alcohol-induced liver injury by regulating the FXR–FGF15 signaling pathway mediated by intestinal microbes and inhibiting lipid and bile acid accumulation in our study (Figure 9).

The changes in mouse liver index and serum ALT, AST, and AST/ALT ratio indicated that this alcohol feeding mode affected the normal growth of mice and damaged their liver cells. However, high–dose LCJ can effectively slow down alcohol–induced liver cell damage. The damage to ALD mouse liver cells may be due to the accumulation of liver lipids or inflammatory cell infiltration caused by alcohol metabolism, or bile acid accumulation and peroxidation [5]. This indicates that the protective effect of LCJ on the liver of ALD mice may be mediated through these aspects.

In our study, we observed that LCJ treatment significantly reduced the increase in serum TG and TC as well as the accretion of fat granules in the liver while increasing the protein expression of AMPK, PPARα, and CPT1b in the liver in ALD mice [37]. AMPK is a key upstream factor that regulates lipid synthesis and catabolism. After AMPK is activated, it can upregulate the expression of PPAR and CPT1. Some scholars believe that AMPK regulates PPAR by activating extracellular signal–regulated kinase (ERK) and the p38 pathway [38]. CPT1 is a rate–limiting enzyme during fatty acid β oxidation, including three subtypes: CPT1A, CPT1B, and CPT1C. PPAR can enhance mitochondrial utilization and the oxidation of fatty acids by promoting the transcriptional expression of CPT1 in the liver. Therefore, we believe that high–dose LCJ can promote the β oxidation of fatty acids by increasing the protein expression of AMPK, PPARα, and CPT1b in the liver of ALD mice, thereby reducing the accumulation of TG and TC in the liver of ALD mice. Studies have also shown that LC extract can actively improve nonalcoholic fatty liver disease by increasing CPT–1, which is involved in fatty acid oxidation, and ACC [5], which supports our conclusion. Polyphenol–rich LC has strong antioxidant activity [39], and dietary polyphenols can activate AMPK in the liver [40]. Hence, LCJ may improve lipid peroxidation in the liver by reducing the production of reactive oxygen species.

Long–term ethanol feeding has been found to reduce intestinal FXR activity. This may be because prolonged alcohol exposure increases FXR acetylation and interferes with FXR’s ability to bind with the retinoid X receptor alpha (RXR), leading to the inactivation of FXR [41]. FXR is a nuclear receptor for bile acids (BAs) and affects the transport and homeostasis of BA. The natural inhibitors of FXR are T–MCA, T–MCA, and UDCA. The strength of the natural activators of FXR follows the order of CDCA > DCA > CA > LCA. GCDCA, TCA, and TDCA are weak FXR activators. Unconjugated BAs have a greater capacity to activate FXR than conjugated BAs [42]. So, the type and amount of BA produced in the intestine may regulate the expression of FXR. Long–term heavy drinking can lead to an imbalance in the gut microbiota that produces bile salt hydrolase (BSH), which is responsible for dehydroxylation reactions. If it leads to a decrease in bile acids with a strong ability to activate FXR in the intestine, it will further promote liver bile acid synthesis by weakening the expression of intestinal FXR. When intestinal FXR activity decreases, it will cause feedback inhibition of the expression of intestinal FGF15, which then increases the expression of the bile acid synthesis gene CYP7A1 in the liver by binding to FGFR4 in the liver, forming a negative feedback loop to increase bile acid synthesis in the liver [43]. This is consistent with the research results of our AG group. However, high-dose LCJ supplementation can increase the expression of FXR, FGF15, and FGFR4 in the ileum and decrease the expression of CYP7A1 to ameliorate liver damage induced by alcohol–induced bile acid accumulation in the liver. Hesperidin is a flavanone glycoside that can prevent cholestatic liver injury and reduce bile acid toxicity in HepaRG cells by activating FXR [44]. Proanthocyanidins can also activate the transcription activity of FXR and reduce triglyceridemia in vivo in an FXR–dependent manner [45]. Chokeberry polyphenols can reduce the relative contents of CA and DCA by altering the composition of the intestinal flora and can increase the relative content of CDCA [46]. Phenolic compounds can be decomposed and utilized by the intestinal flora, which can regulate the intestinal flora. Therefore, we speculate that LCJ rich in phenols may directly activate the activity of intestinal FXR, or it may also activate FXR activity by regulating the content of unconjugated bile acids by regulating the intestinal flora involved in bile acid metabolism. Thus, it alleviates cholestasis by regulating the FXR-FGF15 axis and reducing the expression of CYP7A1 in the liver.

Due to the fact that more than 90% of alcohol is absorbed in the gut after ingestion, excessive alcohol consumption can cause damage to the intestinal barrier. Intestinal barrier damage increases intestinal permeability, which allows the blood to carry LPS, a byproduct of bacteria found in the intestine, into the liver [27]. After LPS is delivered to the liver through the portal vein, as a central mediator of inflammation, it can activate Kupffer cells and macrophages in the liver, which in turn forces these cells to produce inflammatory cytokines such as TNF–α, IL–6, or ROS [1]. HE staining and mucin AB–PAS staining of intestinal sections revealed that alcohol consumption would have a more detrimental effect on the small intestine’s barrier and mucin content than it would on the large intestine. This may be because alcohol is mainly absorbed and metabolized in the small intestine, so the impact on the large intestine is less [7]. Interestingly, high–dose LCJ has the potential to restore alcohol-induced small intestinal barrier damage. Supplementation of LCJ significantly improved alcohol–induced intestinal barrier dysfunction, including the increase of MUC2, MUC4, ZO–1, claudin–1, and the decrease of serum endotoxin levels. These findings supported the findings of the intestinal pathological section staining. These findings imply that supplementing LCJ to improve intestinal barrier function impairments may be an effective way to treat alcohol–induced liver damage.

A high dose of LCJ had different effects on the microbial composition between the large and small intestines. Notably, LCJ increased the relative abundance of *Verrucomicrobia* and *Akkermansia* in the large intestine of ALD mice but decreased the relative abundance of *Parabacteroides*, *Oscillospira*, and *Proteus*. In the small intestine of ALD mice, LCJ treatment increased the relative abundance of *Firmicutes*, *Bacteroidetes*, *Allobaculum*, and *Akkermansia* while reducing the relative abundance of *Proteobacteria*, *Staphylococcus*, and *Proteus*. A previous study reported that *Proteobacteria* contributes to inflammation [47]. In our study, the abundance of Gram–negative bacteria, *Proteobacteria,* was reduced in the HG group, which was consistent with the reduction in the Gram–negative bacteria product LPS. Furthermore, impairment of the intestinal barrier increases the likelihood of LPS entering the bloodstream, while LCJ has been shown to enhance intestinal barrier function. This suggests that LCJ can improve the intestinal barrier and regulate intestinal flora to decrease the production of LPS, thereby reducing blood LPS to improve alcohol-induced liver injury.

*Akkermansia* [48] and *Allobaculum* [49] play important roles in the prevention of alcohol–induced liver damage by increasing intestinal mucus and enhancing gut barrier function. In our study, results showed that the relative richness of *Akkermansia* and *Allobaculum*, the mucin content, and intestinal barrier function in the small intestine of the AG group were significantly reduced, but treatment with LCJ reversed these changes. Moreover, the cecal levels of SCFAs, including acetic acid, propionic acid, butyric acid, and valeric acid, significantly increased in mice with ALD following LCJ intervention. Notably, acetic acid is the most abundant SCFA produced by the gut microbiota and has been shown to improve the function of the intestinal mucosal immune barrier [50]. Butyrate enhances intestinal barrier function by increasing colon mucin and the production of compact junction protein to protect the intestinal mucosal cells from LPS–induced damage [51]. These findings support our results that acetic acid and butyric acid levels in the AG and HG groups were consistent with changes in intestinal compact junction protein levels. *Akkermansia* can produce propionic acid by increasing mucin degradation [52]. *Allobaculum* is physiologically capable of generating butyric acid from carbohydrates [53]. In conclusion, we think that LCJ can increase the abundance of intestinal SCAF–producing bacteria (*Akkermansia* and *Allobaculum*), resulting in high intestinal SCFA production. In this way, it strengthens the intestinal barrier by increasing the expression of intestinal compact junction protein and mucin content to alleviate alcohol–induced liver injury. *Oscillospira* was strongly correlated with the proportion of secondary bile acids in the stool, suggesting that it may contribute to the formation of secondary bile acids in the stool [54]. In the study by Sun et al., it was reported that *Parabacteroides* hydrolyzes a variety of binding cholic acids into secondary cholic acids (stone cholic acid, ursodeoxycholic acid, etc.) through multiple pathways such as bile brine hydrolysase (BSH) [55]. In light of this and the findings of our experiments, we postulate that LCJ, which is abundant in powerful substances like anthocyanins, flavonoids, phenolic acids, and other active polyphenols, can regulate bile acid content as well as FXR and CYP7A1 expression levels in the HG group by improving the abundance of *Oscillospira* and *Parabacteroides* in the intestine. These results demonstrate that LCJ can reduce liver bile acid levels and further ameliorate alcohol-induced liver injury by regulating gut flora via the FXR–FGF15 axis. 

Combined with this study, human treatment of ALD needs to pay attention not only to how to reduce the damage to liver cells but also to the changes in intestinal flora and the intestinal barrier. Plant extracts and probiotics with the impact of enhancing the intestinal barrier and improving intestinal flora are substances worthy of further investigation in the treatment of ALD in the future. This is because ALD can also be relieved by improving the intestinal flora. Additionally, LCJ has no side effects, which helps to maintain a healthy intestinal structure and qualifies it as an ingredient in functional foods for relieving alcoholic liver disease. It also raises public awareness of the health advantages and economic value of LC.

## 5. Conclusions

In conclusion, LCJ can reduce the accumulation of lipids and bile acids in the liver by reducing the levels of serum AST and ALT, thus producing hepatoprotective effects. Moreover, LCJ significantly improved the harm to the intestinal barrier caused by alcohol as well as the imbalance of short–chain fatty acids in the cecum. LCJ was found to reduce alcohol–induced liver injury by regulating gut microbes and its mediated FXR–FGF15 signaling pathway, based on intestinal flora analysis and related gene expression measurements.

## Figures and Tables

**Figure 1 nutrients-15-04025-f001:**
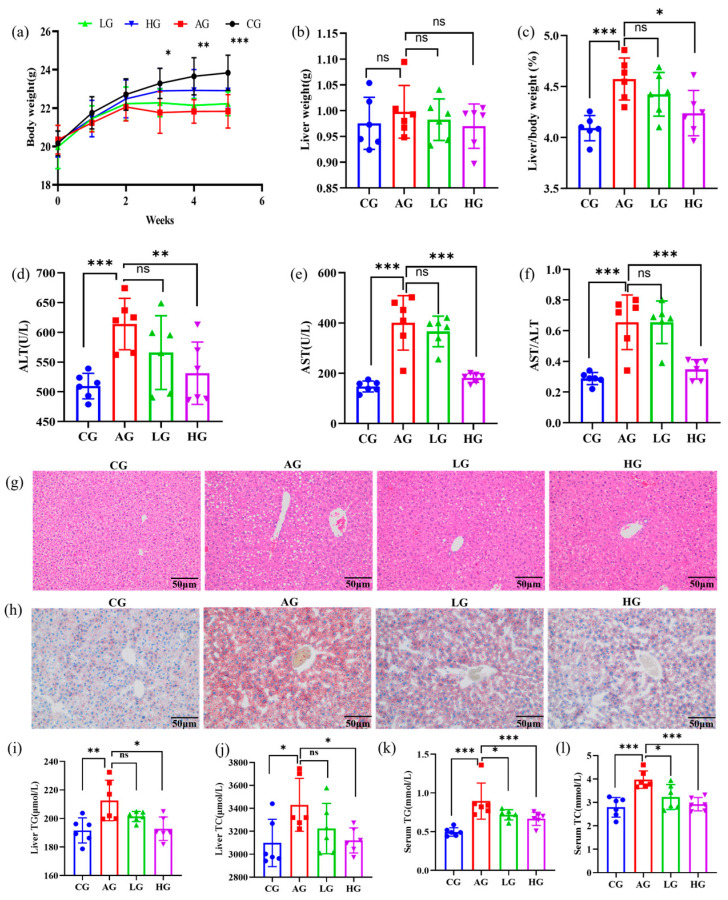
Effects of LCJ intervention on body weight, liver weight, liver index, and lipid accumulation in mice. (**a**) Changes in body weight from week 0 to week 5; (**b**) liver weight; (**c**) liver to body weight ratio; (**d**) serum alanine aminotransferase (ALT); (**e**) serum aspartate aminotransferase (AST); (**f**) ratio of AST/ALT; (**g**) HE staining (200 primary magnification); (**h**) oil red O staining (200 primary magnification); (**i**) liver triglycerides (TG); (**j**) liver cholesterol (TC); (**k**) serum TG; (**l**) serum TC; *n* = 6. ^ns^ *p* > 0.05, * *p* < 0.05, ** *p* < 0.01, and *** *p* < 0.001 vs. the model group (ANOVA).

**Figure 2 nutrients-15-04025-f002:**
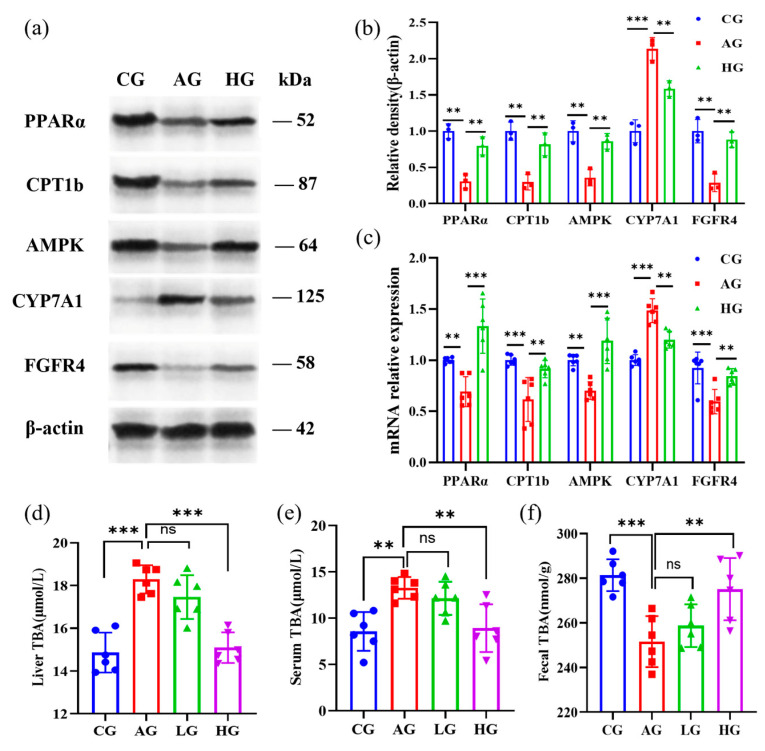
Effects of LCJ on bile acid content, proteins involved in bile acid metabolism, and lipid metabolism in the liver. (**a**,**b**) Protein expression of PPARα, CPT1b, AMPK, CYP7A1, and FGFR4 in the liver of mice (*n* = 3); (**c**) mRNA levels of PPARα, CPT1b, AMPK, CYP7A1, and FGFR4; (**d**) liver TBA; (**e**) serum TBA; (**f**) feces TBA (*n* = 6). ^ns^ *p* > 0.05, ** *p* < 0.01, and *** *p* < 0.001 vs. the model group (ANOVA).

**Figure 3 nutrients-15-04025-f003:**
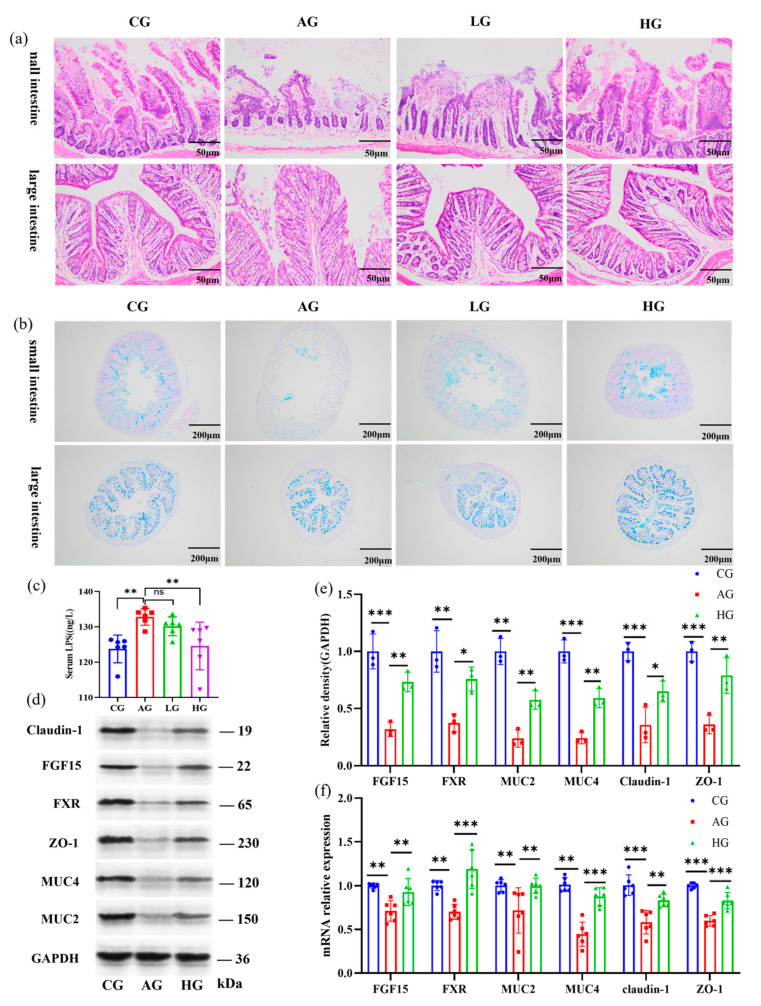
Effects of LCJ on intestinal histological characteristics and the intestinal barrier in mice. (**a**) Hematoxylin and eosin staining (200 primary magnification); (**b**) periodic acid–Schiff and Alcian blue (AB-PAS) staining (50 primary magnification); (**c**) serum LPS; (**d**,**e**) protein expression of FGF15, FXR, MUC2, MUC4, ZO–1, and Claudin–1 in the small intestine (*n* = 3); (**f**) mRNA levels of FGF15, FXR, MUC2, MUC4, ZO–1, and Claudin–1 (*n* = 6). ^ns^ *p* > 0.05, * *p* < 0.05, ** *p* < 0.01, and *** *p* < 0.001 vs. the model group (ANOVA).

**Figure 4 nutrients-15-04025-f004:**
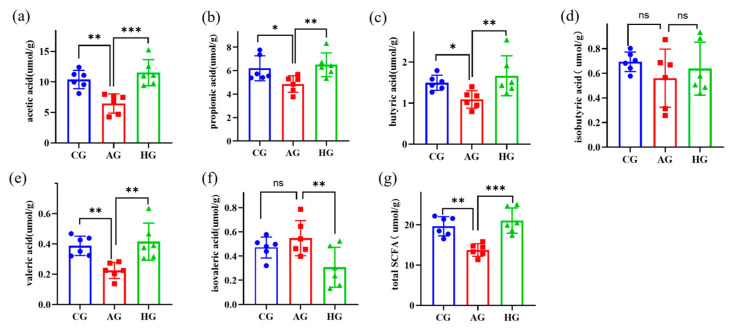
Changes in SCFA levels in the cecum contents of mice. (**a**) Acetic acid; (**b**) propionic acid; (**c**) butyric acid; (**d**) isobutyric acid; (**e**) valeric acid; (**f**) isovaleric acid; (**g**) total SCFAs. *n* = 6. ^ns^ *p* > 0.05, * *p* < 0.05, ** *p* < 0.01, and *** *p* < 0.001 vs. the model group (ANOVA).

**Figure 5 nutrients-15-04025-f005:**
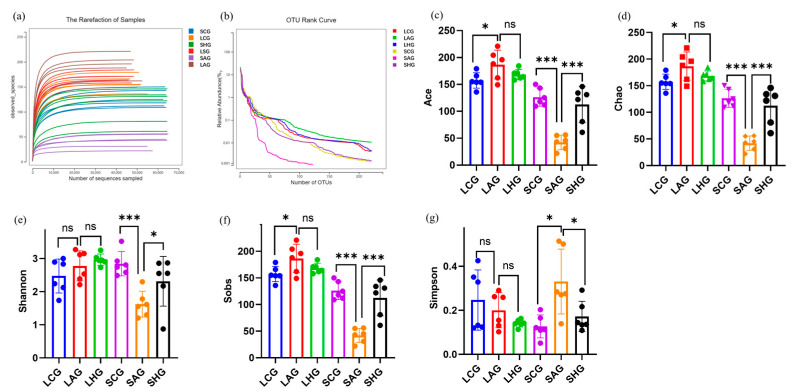
Effects of LCJ on abundance and homogeneity of small intestinal and large intestinal microflora. (**a**) Rarefaction of samples; (**b**) OTU rank curve; (**c**) Ace index; (**d**) Chao index; (**e**) Shannon index; (**f**) Sobs index; (**g**) Simpson index. *n* = 6. ^ns^ *p* > 0.05, * *p* < 0.05 and *** *p* < 0.001 vs. the model group (ANOVA).

**Figure 6 nutrients-15-04025-f006:**
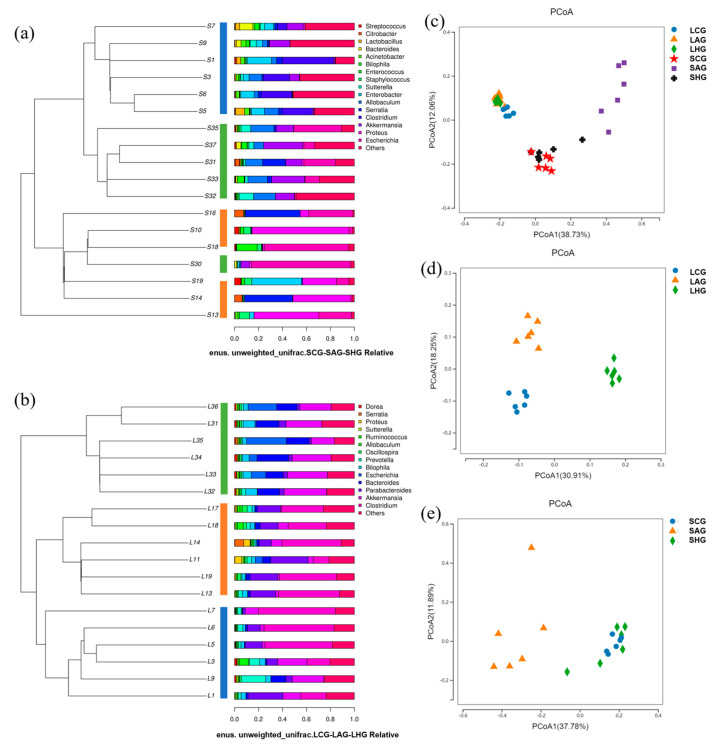
LCJ intervention caused inconsistent changes in gut microbial structure. (**a**) Unweighted pair–group method with arithmetic means cluster tree (small intestine); (**b**) unweighted pair–group method with arithmetic means cluster tree (large intestine); (**c**) principal coordinate analysis (large intestine and small intestine); (**d**) principal coordinate analysis (large intestine); (**e**) principal coordinate analysis (small intestine); *n* = 6.

**Figure 7 nutrients-15-04025-f007:**
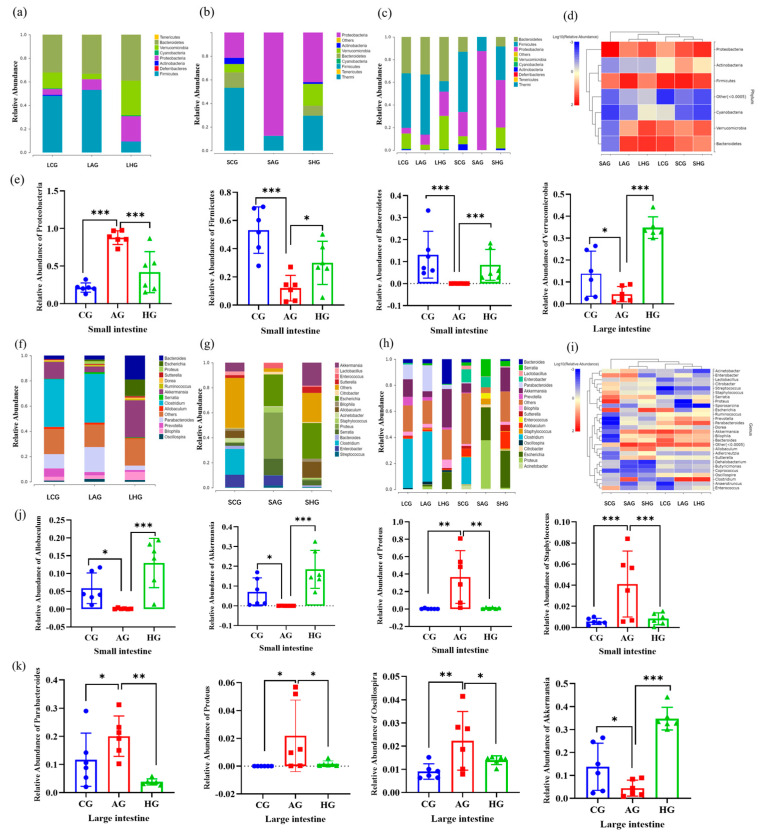
LCJ supplementation altered the composition of microflora in the large and small intestines of ALD mice. (**a**) Large intestine (phylum level); (**b**) Small intestine (phylum level); (**c**) Large intestine and small intestine (phylum level); (**d**) Species composition heat map (phylum level); (**e**) Relative abundance of *Proteobacteria, Firmicutes*, *Bacteroidetes*, and *Verrucomicrobia* (phylum level); (**f**) Large intestine (genus level); (**g**) Small intestine (genus level); (**h**) Large and small intestine (genus level); (**i**) Species composition heat map (genus level); (**j**) Relative abundance of *Allobaculum*, *Akkermansia*, *Proteus*, and *Staphylococcus* (genus level); (**k**) Relative abundance of *Parabacteroides*, *Proteus*, *Oscillospira*, and *Akkermansia* (genus level). *n* = 6. * *p* < 0.05, ** *p* < 0.01, and *** *p* < 0.001 vs. the model group (ANOVA).

**Figure 8 nutrients-15-04025-f008:**
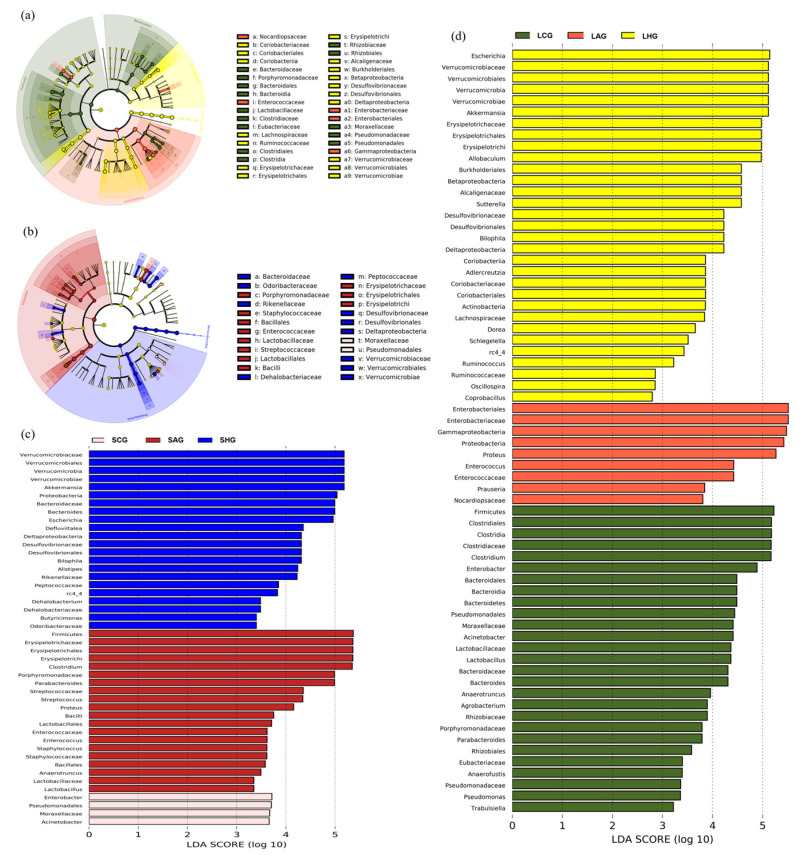
Effect of LCJ on microbial markers (genus level) and metabolic functional pathways. (**a**) Linear discriminant analysis effect size (LEfSe) analysis (small intestine); (**b**) linear discriminant analysis effect size (LEfSe) analysis (large intestine); (**c**) linear discriminant analysis (LDA) in the small intestine; (**d**) linear discriminant analysis (LDA) in the large intestine. *n* = 6.

**Figure 9 nutrients-15-04025-f009:**
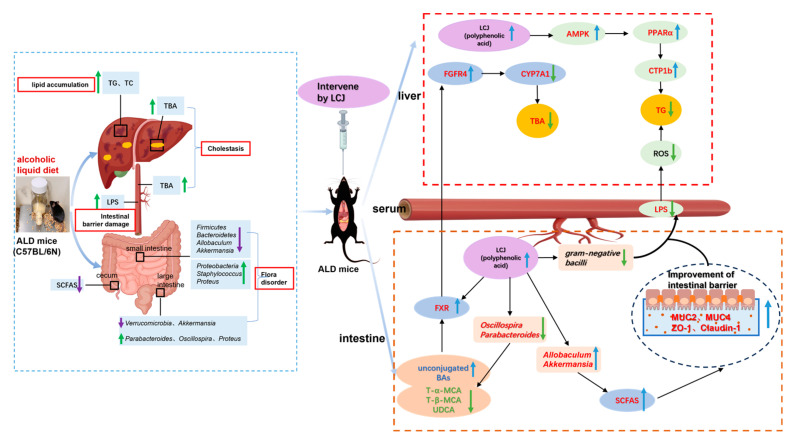
Combination of existing research conclusions and the results of this study. Potential protective mechanism of LCJ against alcohol-induced intestinal injury (Appendix A). The upward arrow represents an increase, while the downward arrow represents a decrease.

## Data Availability

The data presented in this study are available in this article.

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
