# Peer review of "Lonicera Caerulea Juice Alleviates Alcoholic Liver Disease by Regulating Intestinal Flora and the FXR-FGF15 Signaling Pathway"

_nutrients, 2023, doi:10.3390/nu15184025_

Round 1
Reviewer 1 Report
The present manuscript describes a study aimed at evaluating the potential therapeutic action as well as the mechanism of activity of a natural compound, the Lonicera Caerulea Juice (LCJ), for counteracting alcohol-derived liver disease (ALD).
The study represents an interesting effort for identifying nutrients with therapeutic power and the results appear to be promising. However, there are various issues that need to be addressed:
· First of all, the overall experimental design needs to be clarified:
-Authors stated to have included 36 female mice for the present study but they did not explain such selection and, in particular, the reasons why they did not include male mice. Indeed sex-related differences in the response to alcohol absorption and metabolism are well known, thus it would have been interesting to evaluate sex-dependent responses to LCJ.
-Authors subdivided the mice into four groups. However, some analyses are only shown for three groups, namely the CG, AG e HG. If the reason is related to the fact that LG did not show any significant results concerning the body parameters and enzyme dosage (Figure 1c-f) as well as the dosage of Total Bile Acid (Figure 2a-c), authors should clearly state this, thus explaining the rationale of this decision.
· Concerning the Methods section, the illustration of sequencing procedure for the identification of microbiota populations appear to be incomplete. In this section, authors should mention the kit they employed for library preparation and indicate quality parameters and run metrics. These data are essential for the definition of high quality data (lines 153-155).
· The Results section needs to be better organized for enhancing its readability and comprehension: the description of results should be conceptually linked to the number of figures. For instance, the Figures 2d-f are previously mentioned with respect to figures 2a-c (lines 258 and 268). Thus, authors should modify the order of pictures within Figure 2 in order to avoid confusion for the reader. This need to be checked for all the Results section.
· The Discussion section should be improved by better discussing the results other than metagenomics ones (such as histopathological, biochemical and gene expression data). Further insights should be provided concerning the mechanistic link among microbial species and the host gene expression (which could be also be modulated by the LCJ compound itself). A novel picture recapitulating the final model could enhance its comprehension. Moreover how these preclinical results could be translated at the human level for therapeutic purposes should be discussed.
· Minor comments:
-The language should be revised throughout the manuscript and the presence of typos need to be checked.
-The authors should explain the definition of the AG acronym for the “model group”.
- The transcript reference (NM) should be added in Supplementary Table 1.
Overall, the study deserves attention for the interesting and potentially useful outcomes. However, as above mentioned, important information need to be included or better clarified. Thus, considering the above illustrated comments, the present manuscript should be reconsidered for publication upon having performed the indicated revisions (major revisions)
Reviewer 2 Report
This is a very interesting manuscript with a huge amount of data. It is often difficult to follow because of the large number of abbreviations. It maybe advisable to re-write all the abbreviations in each of the figure legends.
Suggest splitting the manuscript into two separate manuscripts and presenting the liver data separately from the intestinal flora data.
Author Response
Response to Reviewer 2 Comments
- Summary
Dear Editor and Reviewers:
On behalf of my co-authors, I would like to thank you for the constructive comments and suggestions on our manuscript entitled “Lonicera Caerulea juice alleviates alcoholic liver disease by regulating intestinal flora and the FXR-FGF15 signaling pathway”.
We have studied your suggestions carefully and made revisions throughout the manuscript. These modifications are highlighted in the revised manuscript, yellow and red stands for revision according to reviewers' opinions. Our responses to the comments from reviewers are provided below (shown in red text).
Thank you again for taking the time to review this manuscript. Please find the detailed responses below and the corresponding revisions.
Point 1: This is a very interesting manuscript with a huge amount of data. It is often difficult to follow because of the large number of abbreviations. It maybe advisable to re-write all the abbreviations in each of the figure legends.
Response 1: Thank you for your suggestions. In order to make readers read more clearly and conveniently, I have compiled an English abbreviation comparison table and added it to the supplementary document. ( This table is also attached at the end of this response.)
Point 2: Suggest splitting the manuscript into two separate manuscripts and presenting the liver data separately from the intestinal flora data.
Response 2: Thank you for your sincere suggestion.
In order to clearly explain the possible mechanism of LCJ in alleviating alcoholic liver disease, we have conducted extensive research, so the data presented may be slightly excessive. In addition, it is also possible that in the initial submission of the manuscript, due to some simple discussions in the results section, the discussion section may not have been elaborated clearly, resulting in a less profound and comprehensive discussion. Therefore, it may make you feel that the connections between the data are not sufficiently relevant, and you have given us such advice. However, based on our research findings, we believe that LCJ alleviates alcoholic liver injury on the one hand because the polyphenolic substances contained in LCJ have the effect of slowing down liver lipid accumulation. In addition, the reduction of bile acids in the liver can also alleviate liver damage caused by alcohol. And we believe that LCJ may directly act on intestinal FXR and may also increase the content of bile acids in the intestine that can activate FXR by improving the gut microbiota related to bile acid metabolism, thereby activating FXR. Importantly, the activation of intestinal FXR can be negatively fed back to the liver, reducing the expression of CYP7A1 in the liver and reducing the production of bile acids. our research revolves around the gut-liver axis, so we did not break down the article into two parts to elaborate. On the basis of this study, perhaps after that, we will focus on the liver or intestines respectively and further explore the mechanisms of LCJ in alleviating alcoholic liver disease. In addition, we not only adjusted the order of the article images, but also rewrote the discussion section of the article and added an additional mechanism diagram (Figure 9) to describe the mechanism of LCJ in alleviating alcoholic liver injury. The highlighted content in yellow and red in the manuscript is the content we have rewritten. I hope the rewritten article can satisfy you.
“Supplementary Table 3. Comparison Table of English Abbreviations”
|
Acronym |
Full Name |
|
ALD |
Alcoholic liver damage |
|
LCJ |
Lonicera caerulea juice |
|
LC |
Lonicera caerulea |
|
TG |
Triglyceride |
|
TC |
Total cholesterol |
|
CG |
Control group |
|
AG |
Alcohol group |
|
LG |
Low-dose LCJ group (total phenol intake: 150 mg/kg) |
|
HG |
High-dose LCJ group (total phenol intake: 300 mg/kg) |
|
SCG |
Small intestine in the control group |
|
SAG |
Small intestine in the alcohol group |
|
SHG |
Small intestine in the high-dose LCJ group |
|
LCG |
Large intestine in the control group |
|
LAG |
Large intestine in the alcohol group |
|
LHG |
Large intestine in the high-dose LCJ group |
|
DADA2 |
Divisive amplicon denoising algorithm |
|
SCFAs |
Short-chain fatty acids |
|
FXR |
Farnesoid X receptor |
|
FGF15 |
Fibroblast growth factor 15 |
|
FGFR4 |
Fibroblast growth factor receptor 4 |
|
CYP7A1 |
Cholesterol 7α-hydroxylase |
|
CPT1b |
Recombinan Carnitine Palmitoyltransferase 1b |
|
LPS |
Lipopolysaccharide |
|
MUC2 |
Mucin2 |
|
ZO-1 |
Zona occludens 1 |
|
AMPK |
Adenosine 5‘-monophosphate (AMP)-activated protein kinase |
|
PPARα |
Peroxisome proliferator-activated receptor |
|
MUC4 |
Mucin4 |
|
TBA |
Total bile acid |
|
qPCR |
Quantitative Real-time PCR |
|
HE |
Hematoxylin-eosin |
|
ALT |
Alanine aminotransferase |
|
AST |
Aspartate aminotransferase |
|
AB-PAS |
Eriodic acid Schiff and Alcian blue |
|
WB |
Western blotting |
|
TβMCA |
Tauro-β-muricholic acid |
|
LCA |
Lithocholic acid |
|
UDCA |
Ursodeoxycholic acid |
|
BSH |
Bile salt hydrolase |
|
CDCA |
Chenodeoxycholic Acid |
|
DCA |
Deoxycholic acid |
|
CA |
Cholic acid |
|
TDCA |
Tauroursodeoxycholic acid |
|
TCA |
Taurocholate acid |
|
GCDCA |
Glycochenodeoxycholic acid |
